# Poison Frogs! Targeted Clean-Label Poisoning Attacks on Neural Networks

**Ali Shafahi**[*]
University of Maryland
ashafahi@cs.umd.edu

**W. Ronny Huang**[*]
University of Maryland
wrhuang@umd.edu

**Mahyar Najibi**
University of Maryland
najibi@cs.umd.edu

**Octavian Suciu**
University of Maryland
osuciu@umiacs.umd.edu

**Christoph Studer**
Cornell University
studer@cornell.edu

**Tudor Dumitras**
University of Maryland
tudor@umiacs.umd.edu

**Tom Goldstein**
University of Maryland
tomg@cs.umd.edu

## Abstract

Data poisoning is an attack on machine learning models wherein the attacker adds examples to the training set to manipulate the behavior of the model at test time. This paper explores poisoning attacks on neural nets. The proposed attacks use "clean-labels"; they don't require the attacker to have any control over the labeling of training data. They are also targeted; they control the behavior of the classifier on a *specific* test instance without degrading overall classifier performance. For example, an attacker could add a seemingly innocuous image (that is properly labeled) to a training set for a face recognition engine, and control the identity of a chosen person at test time. Because the attacker does not need to control the labeling function, poisons could be entered into the training set simply by leaving them on the web and waiting for them to be scraped by a data collection bot.

We present an optimization-based method for crafting poisons, and show that just one single poison image can control classifier behavior when transfer learning is used. For full end-to-end training, we present a "watermarking" strategy that makes poisoning reliable using multiple ($\approx 50$) poisoned training instances. We demonstrate our method by generating poisoned frog images from the CIFAR dataset and using them to manipulate image classifiers.

## 1 Introduction

Before deep learning algorithms can be deployed in high stakes, security-critical applications, their robustness against adversarial attacks must be put to the test. The existence of adversarial examples in deep neural networks (DNNs) has triggered debates on how secure these classifiers are [Szegedy et al., 2013, Goodfellow et al., 2015, Biggio et al., 2013]. Adversarial examples fall within a category of attacks called *evasion attacks*. Evasion attacks happen at test time – a clean target instance is modified to avoid detection by a classifier, or spur misclassification. However, these attacks do not map to certain realistic scenarios in which the attacker cannot control test time data. For example, consider a retailer aiming to mark a competitor's email as spam through an ML-based spam filter. Evasion attacks are not applicable because the attacker cannot modify the victim emails. Similarly,

---

[*]Authors contributed equally.

an adversary may not be able to alter the input to a face recognition engine that operates under supervised conditions, such as a staffed security desk or building entrance. Such systems are still susceptible to *data poisoning* attacks. These attacks happen at training time; they aim to manipulate the performance of a system by inserting carefully constructed *poison instances* into the training data.

This paper studies poisoning attacks on neural nets that are *targeted*, meaning they aim to control the behavior of a classifier on one specific test instance. For example, they manipulate a face recognition engine to change the identity of one specific person, or manipulate a spam filter to allow/deny a specific email of the attacker's choosing. We propose *clean label* attacks that do not require control over the labeling function; the poisoned training data appear to be labeled correctly according to an expert observer. This makes the attacks not only difficult to detect, but opens the door for attackers to succeed without any inside access to the data collection/labeling process. For example, an adversary could place poisoned images online and wait for them to be scraped by a bot that collects data from the web. The retailer described above could contribute to a spam filter dataset simply by emailing people inside an organization.

## 1.1 Related work

Classical poisoning attacks indiscriminately degrade test accuracy rather than targeting specific examples, making them easy to detect. While there are studies related to poisoning attacks on support vector machines [Biggio et al., 2012] or Bayesian classifiers [Nelson et al., 2008], poisoning attacks on Deep Neural Networks (*DNN*) have been rarely studied. In the few existing studies, DNNs have been shown to fail catastrophically against data poisoning attacks. Steinhardt et al. [2017] reported that, even under strong defenses, there is an 11% reduction in test accuracy when the attacker is allowed 3% training set modifications. Muñoz-González et al. [2017] propose a back-gradient based approach for generating poisons. To speed up the process of generating poisoning instances, Yang et al. [2017] develop a generator that produces poisons.

A more dangerous approach is for the attacker to target specific test instances. For example, the retailer mentioned above, besides achieving her target goal, does not want to render the spam filter useless or tip off the victim to the presence of her attack. Targeted backdoor attacks [Chen et al., 2017] with few resources ($\sim$50 training examples) have been recently shown to cause the classifier to fail for special test examples. Gu et al. [2017] trains a network using mislabeled images tagged with a special pattern, causing the classifier to learn the association between the pattern and the class label. In Liu et al. [2017] a network is trained to respond to a trojan trigger.

These attacks present the same shortcomings as evasion attacks; they require test-time instances to be modified to trigger the mispredictions. Moreover, in most prior work, the attacker is assumed to have some degree of control over the labeling process for instances in the training set. This inadvertently excludes real-world scenarios where the training set is audited by human reviewers who will label each example as it appears to the eye, or where the labels are assigned by an external process (such as malware detectors which often collect ground truth labeled by third party antiviruses). Assumed control over the labeling function leads to a straightforward one-shot attack wherein the target instance with a flipped label is added as poison. Overfitting on the poison would then ensure that the target instance would get misclassified during inference time. The most closely related work to our own is by Suciu et al. [2018], who studies targeted attacks on neural nets. This attack, however, requires that poisons fill at least 12.5% (and up to 100%) of *every* minibatch, which may be unrealistic in practice. In contrast, our attacks do not require any control of the minibatching process, and assume a much smaller poisoning budget (<0.1% vs. >12.5%).

Finally, we note that several works have approached poisoning from a theoretical perspective. Mahloujifar and Mahmoody [2017], Mahloujifar et al. [2017] study poisoning threat models from a theoretical perspective, and the robustness of classifiers to training data perturbations was considered in Diakonikolas et al. [2016].

## 1.2 Contributions

In this work, we study a new type of attack, henceforth called *clean-label* attacks, wherein the attacker's injected training examples are cleanly labeled by a certified authority, as opposed to maliciously labeled by the attacker herself. Our strategy assumes that the attacker has no knowledge of the training data but has knowledge of the model and its parameters. This is a reasonable

assumption given that many classic networks pre-trained on standard datasets, such as ResNet [He et al., 2015] or Inception [Szegedy et al., 2014] trained on ImageNet, are frequently used as feature extractors. The attacker's goal is to cause the retrained network to misclassify a special test instance from one class (e.g. a piece of malware) as another class of her choice (e.g. benign application) after the network has been retrained on the augmented data set that includes poison instances. Besides the intended misprediction on the target, the performance degradation on the victim classifier is not noticeable. This makes state-of-the-art poisoning defenses that measure the performance impact of training instances (such as Barreno et al. [2010]) ineffective.

A similar type of attack was demonstrated using influence functions (Koh and Liang [2017]) for the scenario where only the final fully connected layer of the network was retrained on the poisoned dataset, with a success rate of 57%.We demonstrate an optimization-based clean-label attack under the *transfer learning* scenario studied by Koh and Liang [2017], but we achieve 100% attack success rate on the same dog-vs-fish classification task. Further, we study – for the first time to our knowledge – clean-label poisoning in the *end-to-end training* scenario where all layers of the network are retrained. Through visualizations, we shed light on why this scenario is much more difficult due to the expressivity of deep networks. Informed by these visualizations, we craft a 50 poison instance attack on a deep network which achieves success rates of up to 60% in the end-to-end training scenario.

## 2 A simple clean-label attack

We now propose an optimization-based procedure for crafting poison instances that, when added to the training data, manipulate the test-time behavior of a classifier. Later, we'll discuss tricks to boost the power of this simple attack.

An attacker first chooses a *target instance* from the test set; a successful poisoning attack causes this target example to be misclassified during test time. Next, the attacker samples a *base instance* from the base class, and makes imperceptible changes to it to craft a *poison instance*; this poison is injected into the training data with the intent of fooling the model into labelling the target instance with the base label at test time. Finally, the model is trained on the poisoned dataset (clean dataset + poison instances). If, at test time, the model mistakes the target instance as being in the base class, then the poisoning attack is considered successful.

### 2.1 Crafting poison data via feature collisions

Let $f(\mathbf{x})$ denote the function that propagates an input $\mathbf{x}$ through the network to the penultimate layer (before the softmax layer). We call the activations of this layer the *feature space* representation of the input since it encodes high-level semantic features. Due to the high complexity and nonlinearity of $f$, it is possible to find an example $\mathbf{x}$ that "collides" with the target in feature space, while simultaneously being close to the base instance $\mathbf{b}$ in input space by computing

$$\mathbf{p} = \operatorname*{argmin}_{\mathbf{x}} \ \|f(\mathbf{x}) - f(\mathbf{t})\|_2^2 + \beta \|\mathbf{x} - \mathbf{b}\|_2^2 \tag{1}$$

The right-most term of Eq. 1 causes the poison instance $\mathbf{p}$ to appear like a base class instance to a human labeler ($\beta$ parameterizes the degree to which this is so) and hence be labeled as such. Meanwhile, the first term of Eq. 1 causes the poison instance to move toward the target instance in feature space and get embedded in the target class distribution. On a clean model, this poison instance would be misclassified as a target. If the model is retrained on the clean data + poison instances, however, the linear decision boundary in feature space is expected to rotate to label the poison instance as if it were in the base class. Since the target instance is nearby, the decision boundary rotation may inadvertently include the target instance in the base class along with the poison instance (note that training strives for correct classification of the poison instance but not the target since the latter is not part of the training set). This allows the unperturbed target instance, which is subsequently misclassified into the base class during test time, to gain a "backdoor" into the base class.

### 2.2 Optimization procedure

Our procedure for performing the optimization in Eq. 1 to obtain $\mathbf{p}$ is shown in Algorithm 1. The algorithm uses a forward-backward-splitting iterative procedure [Goldstein et al., 2014]. The first (forward) step is simply a gradient descent update to minimize the L2 distance to the target instance

in feature space. The second (backward step) is a proximal update that minimizes the Frobenius distance from the base instance in input space. The coefficient $\beta$ is tuned to make the poison instance look realistic in input space, enough to fool an unsuspecting human observer into thinking the attack vector image has not been tampered with.

---

**Algorithm 1** Poisoning Example Generation

---

**Input:** target instance $t$, base instance $b$, learning rate $\lambda$
Initialize x: $x_0 \leftarrow b$
Define: $L_p(x) = \|f(\mathbf{x}) - f(\mathbf{t})\|^2$
**for** $i = 1$ **to** $maxIters$ **do**
    Forward step: $\widehat{x_i} = x_{i-1} - \lambda \nabla_x L_p(x_{i-1})$
    Backward step: $x_i = (\widehat{x_i} + \lambda \beta b)/(1 + \beta \lambda)$
**end for**

---

## 3 Poisoning attacks on transfer learning

We begin by examining the case of transfer learning, in which a pre-trained feature extraction network is used, and only the final network (softmax) layer is trained to adapt the network to a specific task. This procedure is common in industry where we want to train a robust classifier on limited data. Poisoning attacks in this case are extremely effective. In Section 4, we generalize these attacks to the case of end-to-end training.

We perform two poisoning experiments. First, we attack a pretrained InceptionV3 [Szegedy et al., 2016] network under the scenario where the weights of all layers excluding the last are frozen. Our network and dataset (ImageNet [Russakovsky et al., 2015] dog-vs-fish) were identical to that of Koh and Liang [2017]. Second, we attack an AlexNet architecture modified for the CIFAR-10 dataset by Krizhevsky and Hinton [2009] under the scenario where all layers are trained.[2]

### 3.1 A one-shot kill attack

We now present a simple poisoning attack on transfer learned networks. In this case, a "one-shot kill" attack is possible; by adding just one poison instance to the training set (that is labeled by a reliable expert), we cause misclassification of the target with 100% success rate. Like in Koh and Liang [2017], we essentially leverage InceptionV3 as a feature extractor and retrain its final fully-connected layer weights to classify between dogs and fish. We select 900 instances from each class in ImageNet as the training data and remove duplicates from the test data that are present in the training data as a pre-processing step[3]. After this, we are left with 1099 test instances (698 test instances for the dog class and 401 test instances for the fish class).

We select both target and base instances from the test set and craft a poison instance using Algorithm 1 with $maxIters = 1000$. Since the images in ImageNet have different dimensions, we calculate $\beta$ for Eq. 1 using $\beta = \beta_0 \cdot 2048^2/(dim_b)^2$ which takes the dimensionality of the base instance $(dim_b)$ and the dimension of InceptionV3's feature space representation layer (2048) into account. We use $\beta_0 = 0.25$ in our experiments. We then add the poison instance to the training data and perform cold-start training (all unfrozen weights initialized to random values). We use the Adam optimizer with learning rate of 0.01 to train the network for 100 epochs.

The experiment is performed 1099 times – each with a different test-set image as the target instance – yielding an attack success rate of 100%. For comparison, the influence function method studied in Koh and Liang [2017] reports a success rate of 57% . The median misclassification confidence was 99.6% (Fig. 1b). Further, the overall test accuracy is hardly affected by the poisoning, dropping by an average of 0.2%, with a worst-case of 0.4%, from the original 99.5% over all experiments. Some sample target instances and their corresponding poison instances are illustrated in Fig. 1a.

Note that it is not generally possible to get 100% success rate on transfer learning tasks. The reason that we are able to get such success rate using InceptionV3 on the dog-vs-fish task is because there

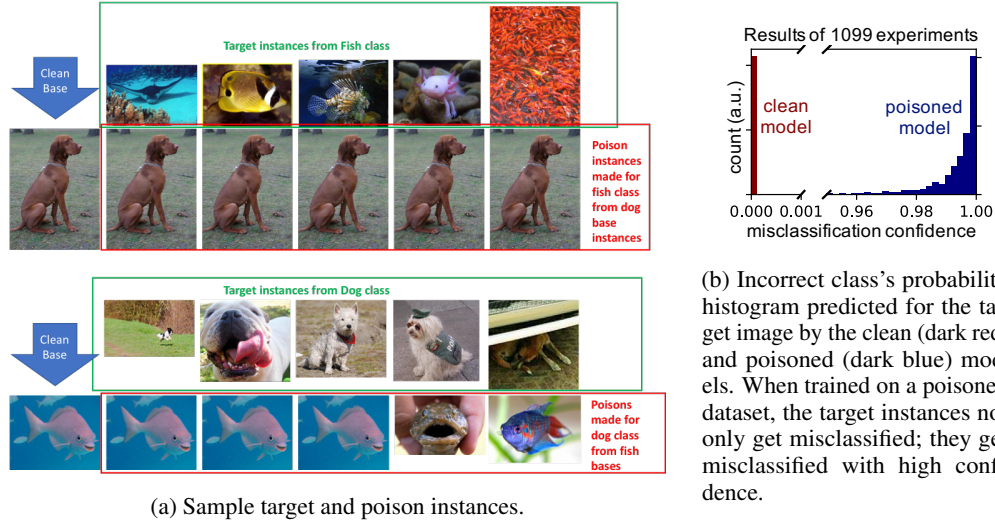

(a) Sample target and poison instances.

(b) Incorrect class's probability histogram predicted for the target image by the clean (dark red) and poisoned (dark blue) models. When trained on a poisoned dataset, the target instances not only get misclassified; they get misclassified with high confidence.

Figure 1: Transfer learning poisoning attack. (a) The top row contains 5 random target instances (from the "fish" class). The second row contains the constructed poison instance corresponding to each of these targets. We used the same base instance (second row, leftmost image) for building each poison instance. The attack is effective for any base, but fewer iterations are required if the base image has a higher resolution. We stopped the poison generation algorithm when the maximum iterations was met or when the feature representation of the target and poison instances were less than 3 units apart (in Euclidean norm). The stopping threshold of 3 was determined by the minimum distance between all pairs of training points. As can be seen, the poison instances are visually indistinguishable from the base instance (and one another). Rows 3 and 4 show samples from similar experiments where the target (fish) and base (dog) classes were swapped.

are more trainable weights (2048) than training examples (1801). As long as the data matrix contains no duplicate images, the system of equations that needs to be solved to find the weight vector is under-determined and overfitting on all of the training data is certain to occur.

To better understand what causes the attacks to be successful, we plot the angular deviation between the decision boundary (i.e. the angular difference between the weight vectors) of the clean and poisoned networks in Fig. 2 (blue bars and lines). The angular deviation is the degree to which retraining on the poison instance caused the decision boundary to rotate to encompass the poison instance within the base region. This deviation occurs mostly in the first epoch as seen in Fig. 2b, suggesting that the attack may succeed even with suboptimal retraining hyperparameters. The final deviation of 23 degrees on average (Fig. 2a) indicates that a substantial alteration to the final layer decision boundary is made by the poison instance. These results verify our intuition that misclassification of the target occurs due to changes in the decision boundary.

While our main formulation (Eq. 1) promotes similarity between the poison and base images via the $\ell_2$ metric, the same success rate of 100% is achieved when we promote similarity via an $\ell_\infty$ bound of 2 (out of a dynamic range of 255) as is done in Koh and Liang [2017]. Details of the experiment are presented in the supplementary material.

The experiments here were on a binary classification task ("dog" vs. "fish"). There is, however, no constraint on applying the same poisoning procedure to many-class problems. In the supplementary material, we present additional experiments where a new class, "cat", is introduced and we show 100% poisoning success on the 3-way task is still achieved while maintaining an accuracy of 96.4% on clean test images.

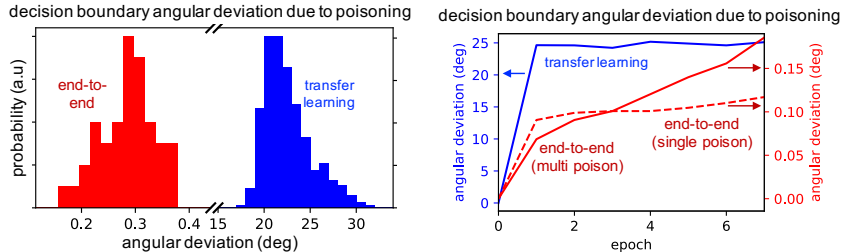

(a) PDF of decision boundary ang. deviation.    (b) Average angular deviation vs epoch.

Figure 2: Angular deviation of the feature space decision boundary when trained with clean dataset + poison instance(s) versus when trained with clean dataset alone. (a) Histogram of the final (last epoch) angular deviation over all experiments. In transfer learning (blue), there is a significant rotation (average of 23 degrees) in the feature space decision boundary. In contrast, in end-to-end training (red) where we inject 50 poison instances, the decision boundary's rotation is negligible. (b) Most of the parameter adjustment is done during the first epoch. For the end-to-end training experiments, the decision boundary barely changes.

# 4   Poisoning attacks on end-to-end training

We saw in Section 3 that poisoning attacks on transfer learning are extremely effective. When all layers are trainable, these attacks become more difficult. However, using a "watermarking" trick and multiple poison instances, we can still effectively poison end-to-end networks.

Our end-to-end experiments focus on a scaled-down AlexNet architecture for the CIFAR-10 dataset[4] (architectural details in appendix), initialized with pretrained weights (warm-start), and optimized with Adam at learning rate $1.85 \times 10^{-5}$ over 10 epochs with batch size 128. Because of the warm-start, the loss was constant over the last few epochs after the network had readjusted to correctly classify the poison instances.

## 4.1   Single poison instance attack

We begin with an illustrative example of attacking a network with a single poison instance. Our goal is to visualize the effect of a poison on the network's behavior, and explain why poisoning attacks under end-to-end training are more difficult than under transfer learning. For the experiments, we randomly selected "airplane" as the target class and "frog" as the base class. For crafting poison instances, we used a $\beta$ value of 0.1 and iteration count of 12000. Figure 3a shows the target, base, and poison feature space representations visualized by projecting the 193-dimensional deep feature vectors onto a 2-dimensional plane. The first dimension is along the vector joining the centroids of the base and target classes ($\mathbf{u} = \mu_{base} - \mu_{target}$), while the second dimension is along the vector orthogonal to $\mathbf{u}$ and in the plane spanned by $\mathbf{u}$ and $\theta$ (the weight vector of the penultimate layer, i.e. the normal to the decision boundary). This projection allows us to visualize the data distribution from a viewpoint best representing the separation of the two classes (target and base).

We then evaluate our poisoning attack by training the model with the clean data + single poison instance. Fig. 3a shows the feature space representations of the target, base, and poison instances along with the training data under a clean (unfilled markers) and poisoned (filled markers) model. In their clean model feature space representations, the target and poison instances are overlapped, indicating that our poison-crafting optimization procedure (Algorithm 1) works. Oddly, unlike the transfer learning scenario where the final layer decision boundary rotates to accommodate the poison instance within the base region, the decision boundary in the end-to-end training scenario is *unchanged* after retraining on the poisoned dataset, as seen through the red bars and lines in Fig. 2.

From this, we make the following important observation: *During retraining with the poison data, the network modifies its lower-level feature extraction kernels in the shallow layers so the poison instance is returned to the base class distribution in the deep layers.*

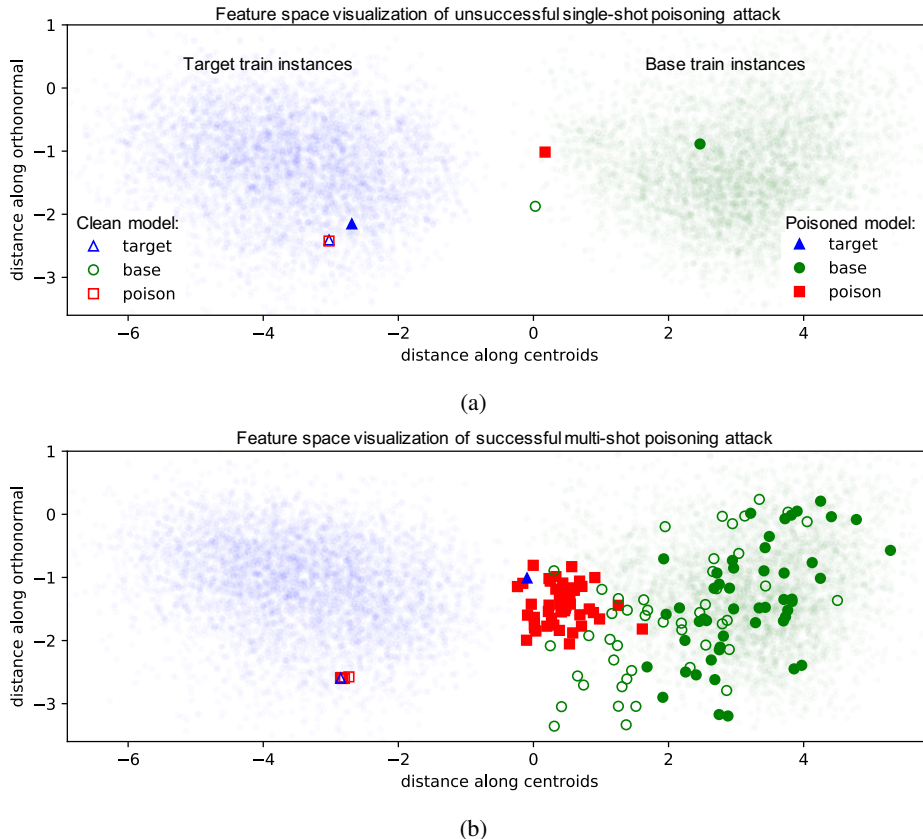

(a)

(b)

Figure 3: Feature space visualization of end-to-end training poisoning attacks. (a) A single poison instance is unable to successfully attack the classifier. The poison instance's feature space position under the clean model is overlapped with that of the target instance. However, when the model is trained on the clean + poisoned data (i.e. the poisoned model), the feature space position of the poison instance is returned to the base class distribution, while target remains in the target class distribution. (b) To make the attack successful, we construct 50 poison instances from 50 random base instances that are "watermarked" with a 30% opacity target instance. This causes the target instance to be pulled out of the target class distribution (in feature space) into the base class distribution and get incorrectly classified as the base class.

In other words, the poison instance generation exploits imperfections in the feature extraction kernels in earlier layers such that the poison instance is placed alongside the target in feature space. When the network is retrained on this poison instance, because it is labeled as a base, those early-layer feature kernel imperfections are corrected and the poison instance is returned to the base class distribution. This result shows that the objectives of poison instance generation and of network training are mutually opposed and thus a single poison may not be enough for compromising even extreme outlier target examples. To make the attack successful, we must find a way to ensure that the target and poison instances do not get separated in feature space upon retraining.

## 4.2 Watermarking: a method to boost the power of poison attacks

To prevent the separation of poison and target during training, we use a simple but effective trick: add a low-opacity watermark of the target instance to the poisoning instance to allow for some inseparable feature overlap while remaining visually distinct. This blends some features of the target instance into the poison instance and should cause the poison instance to remain within feature space proximity of the target instance even after retraining. Watermarking has been previously used in Chen et al. [2017], but their work required the watermark to be applied during inference time, which is unrealistic in situations where the attacker cannot control the target instance.

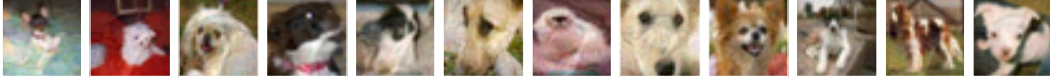

Figure 4: 12 out of 60 random poison instances that successfully cause a bird target instance to get misclassified as a dog in the end-to-end training scenario. An adversarial watermark (opacity 30%) of the target bird instance is applied to the base instances when making the poisons. More examples are in the supplementary material.

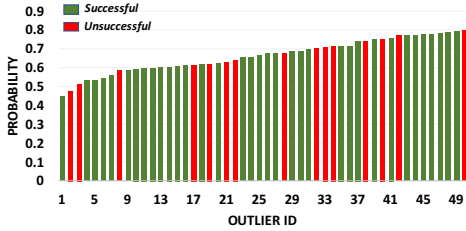

(a) Attacks on the most outlier target airplanes. The bars indicate the probability of the target instance before the attack (calculated using the pre-trained network). The coloring, denotes whether the attack was successful or unsuccessful. Each experiment utilizes a watermark opacity of 30% and 50 poisons. Out of these 50 outliers, the attack succeeds 70% of the time (compare to 53% for a random target).

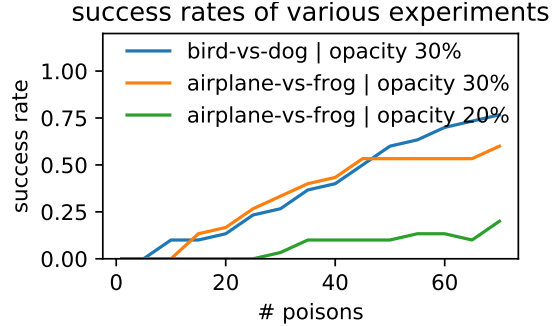

(b) Success rate of attacks on different targets from different bases as a function of number of poison instances used and different target opacity added to the base instances.

Figure 5: Success rates for attacks on outliers and random targets. While attacking non-outlier is still possible, attacking an outlier can increase the chances of success.

A base watermarked image with target opacity $\gamma$ is formed by taking a weighted combination of the base $b$ and the target images $t$: $\mathbf{b} \leftarrow \gamma \cdot \mathbf{t} + (1 - \gamma) \cdot \mathbf{b}$. Some randomly selected poison instances are shown in the supplementary material. Watermarks are not visually noticeable even up to 30% opacity for some target instances. Fig. 4 illustrates 60 poison instances used for successfully attacking a "bird" target instance.

### 4.2.1 Multiple poison instance attacks

Poisoning in the end-to-end training scenario is difficult because the network learns feature embeddings that optimally distinguish the target from the poison. But what if we introduce multiple poison instances derived from different base instances into the training set?

For the classifier to resist multiple poisons, it must learn a feature embedding that separates *all* poison instances from the target while also ensuring that the target instance remains in the target distribution. We show in our experiments that using a high diversity of bases prevents the moderately-sized network from learning features of the target that are distinct from those of the bases. Consequently, when the network is retrained, the target instance is pulled along with the poison instances toward the base distribution, and attacks are frequently successful. These dynamics are shown in Fig. 3b.

In Fig. 2, we observe that even in the multiple poison experiments, the decision boundary of the final layer remains unchanged, suggesting that there's a fundamentally different mechanism by which poisoning succeeds in the transfer learning vs. end-to-end training scenarios. Transfer learning reacts to poisons by rotating the decision boundary to encompass the target, while end-to-end training reacts by pulling the target into the base distribution (in feature space). The decision boundary in the end-to-end scenario remains stationary (varying by fractions of a degree) under retraining on the poisoned dataset, as shown in Fig. 2.

To quantify how the number of poison instances impacts success rate, we ran experiments for each number of poison instances between 1 and 70 (increments of 5). Experiments used randomly chosen target instances from the test set. Each poison was generated from a random base in the test set (resulting in large feature diversity among poisons). A watermarking opacity of 30% or 20% was used to enhance feature overlap between the poisons and targets. The attack success rate (over 30 random trials) is shown in Fig. 5b. The set of 30 experiments was repeated for a different target-base class pair within CIFAR-10 to verify that the success rates are not class dependent. We also try a

lower opacity and observe that the success rate drops. The success rate increases monotonically with the number of poison instances. With 50 poisons the success rate is about 60% for the bird-vs-dog task. Note we declare success only when the target is classified as a base; the attack is considered unsuccessful even when the target instance is misclassified to a class other than the base.

We can increase the success rate of this attack by targeting data outliers. These targets lie far from other training samples in their class, and so it should be easier to flip their class label. We target the 50 "airplanes" with the lowest classification confidence (but still correctly classified), and attack them using 50 poison frogs per attack. The success rate for this attack is 70% (Fig. 5a), which is 17% higher than for randomly chosen targets.

To summarize, clean-label attacks under the end-to-end scenario require multiple techniques to work: (1) optimization via Algorithm 1, (2) diversity of poison instances, and (3) watermarking. In the supplementary material, we provide a leave-one-out ablation study with 50 poisons which verifies that all three techniques are required for successful poisoning.

## 5    Conclusion

We studied targeted clean-label poisoning methods that attack a net at training time with the goal of manipulating test-time behavior. These attacks are difficult to detect because they involve non-suspicious (correctly labeled) training data, and do not degrade the performance on non-targeted examples. The proposed attack crafts poison images that collide with a target image in feature space, thus making it difficult for a network to discern between the two. These attacks are extremely powerful in the transfer learning scenario, and can be made powerful in more general contexts by using multiple poison images and a watermarking trick.

Training with poison instances is akin to the *adversarial training* technique for defending against evasion attacks (Goodfellow et al. [2015]). The poison instance can here be seen as an adversarial example to the base class. While our poisoned dataset training does indeed make the network more robust to base-class adversarial examples designed to be misclassified as the target, it also has the effect of causing the unaltered target instance to be misclassified as a base. This side effect of adversarial training was exploited in this paper, and is worth further investigation.

Many neural networks are trained using data sources that are easily manipulated by adversaries.We hope that this work will raise attention for the important issue of data reliability and provenance.

## 6    Acknowledgements

Goldstein and Shafahi were supported by the Office of Naval Research (N00014-17-1-2078), DARPA Lifelong Learning Machines (FA8650-18-2-7833), the DARPA YFA program (D18AP00055), and the Sloan Foundation. Studer was supported in part by Xilinx, Inc. and by the US National Science Foundation (NSF) under grants ECCS-1408006, CCF-1535897, CCF-1652065, CNS-1717559, and ECCS-1824379. Dumitras and Suciu were supported by the Department of Defense.

## Footnotes

[2]The code is available at `https://github.com/ashafahi/inceptionv3-transferLearn-poison`

[3]If an identical image appears in both the train and test set, it could be chosen as both a base and target, in which case poisoning is trivial. We remove duplicate images to prevent this sort of "cheating."

[4]We do this to keep runtimes short since quantifying performance of these attacks requires running each experiment (and retraining the whole network) hundreds of times.

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
