[Supplementary Material]

# A Illustrations of Poisoning and how the decision boundary will rotate after training with the poison instances

1a illustrates our poisoning scheme. When a network is trained on clean data + the very few poison instance, the linear decision boundary in feature space (Figure 1b) is expected to rotate to include the poison instance in the base class side of the decision boundary in order to avoid misclassification of that instance.

(b) Illustration of the feature space (activations of the penultimate layer before the softmax layer of the network) representation of clean training data, poison instance, and target instance. Note that the target instance is *not* in the training set. Thus it will not affect the loss when the nearby poison instance causes the decision boundary to shift to encompass both of them into the base region.

Figure 1: (a) Schematic of the clean-label poisoning attack. (b) Schematic of how a successful attack might work by shifting the decision boundary.

# B Comparison to adversarial examples

The clean-label targeted poison attack is similar to adversarial examples in the sense that they both are used for misclassifying a particular target instance. However, they differ in the kinds of freedom the attacker has on manipulating the target. Adversarial examples assume that the target can be slightly modified and hence they craft an example which looks very similar to the target instance in input space but gets misclassified. In the targeted clean-label framework, we assume that the attacker has no control over the target instance during test time and can not (or is not willing to) modify it even slightly. This makes the threat posed more concerning, as it allows one to control the classifier's decisions on test-time instances outside their spectrum of control. This framework could be also useful for fooling a face recognition system. While it has been shown that an adversary can craft accessories for a target such that wearing that accessory causes the face-recognition system to misclassify the target, there are many sensitive situations which the target is prevented from wearing any accessories.

# C Pixel bounded (i.e. $L_\infty$) poisoning attacks

In the main body, the optimization formulation enforces input image similarity between the poison and base image by adding an $L_2$ penalty. Different similarity measures can be enforced by changing this. For example, we can enforce similarity measured using $L_\infty$ by solving the following optimization problem:

$$\mathbf{p} = \underset{\mathbf{x}}{\text{argmin}} \quad \|f(\mathbf{x}) - f(\mathbf{t})\|_2^2 \tag{1}$$
$$s.t. \quad \|\mathbf{x} - \mathbf{b}\|_\infty \leq \epsilon_\infty$$

This optimization problem can be effectively solved by performing clipping after each gradient descent update during each iteration. We clip every pixel of the poison image to be within the $\epsilon_\infty$ of the base image. To be consistent with the experiments of Koh and Liang [2017], we use $\epsilon_\infty = 2$. Our "one-shot kill" attack maintains its 100% success rate under the $L_\infty$ bounded poisonings attacks.

## D   One-shot kill attacks for multi-class transfer-learning scenario

In the main body, the first experiment we performed was transfer learning for a binary classification task where the two classes were dogs, and fish. While our successful attack experiments for the transfer learning part were on a binary classification task, they generalize to multi-class problems as long as the number of training data is less than the number of parameters – ensuring that it is possible to over-fit on the poison instance. Note that the data matrix, propagated to feature space, is $N_{tr}$ (examples) $\times n_d$ (features), which in the transfer learning scenario setting is underdetermined, implying that there are multiple solutions (weights) which allow perfect classification of all examples. The problem is underdetermined since the network in this situation has $n_d \times C$ trainable parameters and $N_{tr}$ equations. Here, $n_d = 2048$ is the deep feature dimensionality of the InceptionV3 model and $C$ is the number of classes. While the math certainly supports overfitting when $N_{tr} \leq n_d \times C$ and there are no duplicates in $N_{tr}$, the optimization would also overfit in practice. The loss function of the one-layer linear classification problem is convex and SGD is guaranteed to find the global minimizer (zero loss, i.e., overfit). Indeed, empirical evidence supports overfitting: training loss is near zero ($<10^{-5}$ after 100 epochs) and all training samples are correctly classified across all experiments. Interestingly, going from a binary classification problem to a multi-class problem can even be advantageous for the adversary as it adds to the number of trainable network parameters by increasing $C$. We experimentally verify this by adding the "cat" class to the already existing fish and dog classes.

Consistent with other experiments, we randomly pick 900 examples as 'clean' training samples from the cat class and augment this to the 1800 examples of the dog and fish classes that we had. Note that here $N_{tr} = 2700$ and $C = 3$. The test accuracy on the clean examples for this case is 96.4%. We achieve 100% attack success rate using our "one-shot kill" attack when we build poisons for 100 randomly sampled targets from the test set.

## E   Sampling the candidate target instance for more success

As mentioned in the main body, a smart attacker will maximize her chances of success by choosing an effective and easy to manipulate target instance. Because outliers are separated from most training samples, the decision boundary can easily change in this location without substantially affecting classification accuracy. Also, points chosen near the decision boundary require less manipulation of the classifier in order to change its behavior. Note that the 2D illustration in Figure 1b belies the relative ease with which this condition can be fulfilled in higher dimensional spaces; nonetheless, this condition provides a heuristic by which we chose our target instances when we want higher success rates.

## F   Network architecture for CIFAR-10 classifier

The scaled down AlexNet architecture attacked during the end-to-end experiments is summarized in Table 1. Without poisoning, the network has a training accuracy of 100% and a test accuracy of 74.5%. To reach this accuracy, the network is trained for 200 epochs on clean data (non-poisoned) with a learning rate that decays with a schedule. The final learning rate is the one used for retraining the model once the training data set is poisoned.

Figure 2: *All* 60 poison instances that successfully cause a bird target instance get misclassified as a dog in the end-to-end training scenario. An adversarial watermark (opacity 30%) of the target bird instance is applied to the base instances used for making the poisons.

Table 1: Network architecture for the poisoning of CIFAR-10 experiments.

|   | Type | Kernel Size | #Out Dim. |
|---|---|---|---|
| 1 | Conv+ReLU | $5 \times 5$ | 64 |
| 2 | MaxPool 1/2 | $3 \times 3$ | 64 |
| 3 | LRN | - | 64 |
| 4 | Conv+ReLU | $5 \times 5$ | 64 |
| 5 | MaxPool 1/2 | $3 \times 3$ | 64 |
| 6 | LRN | - | 64 |
| 7 | FullyConnected+ReLU | - | 384 |
| 8 | FullyConnected+ReLU | - | 192 |
| 9 | FullyConnected | - | 10 |

## G   What do watermarked poisons look like?

The poison instances' appearance depends on the target instance being attacked and also the level of transparency (opacity) of the target instance being added to the base instances for making poison instances (Fig. 3). Attacking target instances from the birds class using poison instances built from base instances belonging to the dog class (Fig. 5) are both more successful and the poison instance images look less disturbed than the airplane-vs-frog attack (Fig. 4). For example, the complete set of 60 poison examples used for successfully attacking a target bird is illustrated in Fig. 2). If the auditor does not know the target instance (similar to the situation in Fig. 2) and is not seeing all of the poison instances side-by-side, the chances that the poison instances would be flagged as threats should not be very high.

When multiple objects are present in am image, they could be exploited to craft an attack. For example, many images of the ImageNet data set contain multiple objects from different classes, although the image has only one label. A clever watermarking for these situations could add the target instance or parts of the target instance with a higher opacity in a place where it seems innocuous. For example, one can add the airplane target instance flying in the background of a poisoned frog instance. Or if the task is to misclassify Bob as Alice, the attacker can add images of Bob to group photos in which Alice is present.

### G.1   Ablation study: How many frogs does it take to poison a network?

We perform a leave-one-out ablation study to show the effect of each of the poisoning methods described above. Feature representations for attacking a particular target instance (before and after

Figure 3: Poison frogs. Every row contains optimized poisoning instances built from random images belonging to the base class (frog) for the given candidate target instance that belongs to the airplane class. We apply an adversarial watermark (a transparent overlay of the target instance "airplane" image) with different opacity levels. These poisoning instances are close to the airplane in feature space. For CIFAR-10 images, 30% opacity watermarks are often hard to recognize and an unassuming human labeler would almost certainly properly label these images as "frog." However, for opacity 50% the watermark is noticeable.

adversarial training) are visualized in Fig. 6. Results are shown for a process using all of the methods described above (row a), which produces a successful attack.

Row (b) shows a process that only uses one base to produce all the poisons. Training with multiple adversarial instances from the same base is called "adversarial training," and makes the network robust against adversarial examples [Goodfellow et al., 2015]. For this reason, reusing the base image causes poisoning to fail. Row (c) shows a process that excludes optimization to produce feature collisions, and (d) shows a process that leaves out watermarking.

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

Figure 4: Some samples of poisons made for different plane targets using base instances as frogs. Note that, the bases were watermarked with a 30% opacity of the target. The target instance is on the top row and some of the poisons are below it. A green dot is used when the attack was successful and red dots indicate unsuccessful attacks. Note that the images do not always look clean compared to the bird vs dog task presented in Fig. 5. But it is important to note that when the target is not known, it may be hard to recognize it's existence in all images.

Figure 5: Some sample target and poison instances for the task of attacking a target instance from the bird class using base instances from the dog class. The columns with the green dot above them are successful attacks. Similar, to the other experiment (attack), a 30% opacity of the target is added to the base instance. Note that the poison instances here are better looking than the poison instances of the plane-frog task. Also, the attacks are more successful for this task. To verify that this attack is hardly detectable to an non-suspicious label auditor, one should only look at one column and ignore the top row because the auditor/labeler is not aware of the target.

(a) Successful attack - includes: optimization + watermarking + multi-base

(b) Unsuccessful attack - includes: optimization + watermarking ; excludes: multi-base

(c) Unsuccessful attack - includes: multi-base + watermarking ; excludes: optimization

(d) Unsuccessful attack - includes: optimization + multi-base ; excludes: watermarking

Figure 6: Ablation study showing the effect of different components of the poisoning attack on a network with end-to-end training. The target is in the "airplane" class, and has roughly 70% class probability of belonging to its home class. Dark green dots are bases, red squares are poisons, and the dark blue triangle is the target. Translucent points are other data points in each class. (a) Two epochs of training on a successful attack that includes watermarking, 50 poison frogs from random bases, and optimization. The base instances are skewed towards the target image because they have 30% opacity of the target. It can be seen that most of the poisoning happens during the first epoch of retraining. The other rows depict unsuccessful attacks. (b) Multiple poisons are used, but all from the same base. (c) Watermarking and multi-base poisons are used, but without optimization to collide the feature representations of the poisons with the target. Unlike the other rows, the base instances (dark green) do not include the watermarking while the poisons do. (d) No watermarking.