[Reviews · NeurIPS 2018]

Reviewer 1



**Post feedback update** I would like to thank the authors for their feedback. Based on the information provided, I would encourage the following changes be made to the paper in addition to what was suggested in the original reviews: * Improve comparison with related work by Suciu et al. and Koh and Liang (using points on the more realistic treat model and also experimental results mentioned in your feedback). * Tone claims down with respect to transfer learning or support with additional empirical evidence. **End post feedback update** This manuscript introduces a poisoning attack that departs from prior work by not requiring that the adversary control the labeling mechanism (the poisoning operation is not what is commonly referred to as “label flipping”). They also don’t require the presence of a trigger at test time (as done in what is commonly referred to as a “backdoor”). To the best of my knowledge, it includes pointers to relevant prior work both on shallow and deep learning algorithms. For these reasons, I think the manuscript is original and should contribute to increased visibility of poisoning in the adversarial ML community. Furthermore, the manuscript is well written and easy to follow. The attacks discussed are intuitive, in particular to readers familiar with the adversarial example literature. Some observations made in the submission, although perhaps simple in retrospect, are interesting to pounder on: for instance, end-to-end training affects how lower layers represent the data in feature space while transfer learning only shifts decision boundaries of the last layer (because that’s the only layer that can be modified). This should shine some light on potential mitigation mechanisms. Perhaps the main limitation of the submission is the strong assumptions it makes about adversarial capabilities in its threat model: the adversary needs to have access to both the model architecture and parameters resulting from training. Another limitation of the present work on poisoning is also found in the adversarial example literature, and thus it would be nice to anticipate it to avoid some of the pitfalls encountered in the context of adversarial examples: L_2 norms are not an idea characterization of how a perturbation affects human perception. Thus, low L_2 norms do not ensure indistinguishability to the human observer. While there are likely no solutions to this problem at this time, it would be useful to include a discussion of the attack’s applicability to different domains (beyond vision), and which metric they would call for. Will it always be possible to define a differentiable constraint on the perturbation (e.g., for emails in the SPAM scenario you discuss)? Finally, I had some concerns about the scalability of the approach. Does the attack require the model to overfit on its training data? Some evidence that controls for overfitting would be beneficial. Is the number of poisoning points high relatively to the number of training points per class? Furthremore, the transfer learning evaluation only considers a binary classification problem. How would it extend to multi-class tasks? Some additional comments that would be nice to clarify in the rebuttal: * Does the base instance need to be part of the original training dataset? * How does the approach scale when the adversary is not targeting a single test input but instead multiple target test inputs? Would the architecture size and potential overfitting issue mentioned above matter more in this setting? * Could you clarify the motivation for the procedure used in Section 2.2? Why not apply a more common optimizer like Adam? Is there something different about the particular loss used? * In practice, is watermarking the most efficient poisoning attack in these settings? Does it consistently outperform the procedure from Section 2.2? * Why is angular deviation relevant in high dimensional settings?

Reviewer 2



== Update after author response == The authors' points about the comparison to Suciu et al. and Koh and Liang are well taken, thanks. I still think that the evidence for the conclusions about end-to-end training vs. transfer learning is weak. While training the Inception network from scratch might be infeasible, the authors should still have done a more controlled experiment, e.g., transfer learning vs. end-to-end training on CIFAR. The claims in the paper about that comparison are currently too speculative. == Original review == The authors propose a method for creating data poisoning attacks that crafts attack points so that they resemble the target point (that the attacker wants to get misclassified) in feature space, while resembling an innocuous base point in input space. The authors run a carefully-designed series of experiments to show that their targeted attack succeeds in both a transfer learning setting as well as a more challenging end-to-end training setting. The paper is well-written and easy to follow, and the authors take pains to do an ablative analysis of their attacks, as well as try to analyze them beyond just noting their effectiveness (e.g., by looking at the rotation of the decision boundary). The attack idea is simple (a good thing), though somewhat specific to neural networks, and seems quite workable in practice (which is bad news for most of us, but a good contribution). My main concern is about novelty, especially compared to the Suciu et al. (2018), which the authors acknowledge. Conceptually, the Stingray attack introduced in Suciu et al. is very similar to the Poison Frogs attack that the present paper introduces: the former also tries to craft poisoned points that are similar in feature space to the attack point but similar in input space to a benign base point. The attacks differ in that Suciu et al. measure similarity in input space in the L_\inf norm (i.e., max deviation of each pixel), which is I think more justified than the L2 norm that the present authors use. Given the comparisons to adversarial training throughout the paper -- where the L_\inf norm is much more commonly used -- the choice of the L2 norm is unorthodox, and I think the onus is on the present authors to defend it. The drawback of Suciu et al. is in their evaluation, because they weave in a constant fraction of the poisoned points into each training minibatch: but this seems to be a flaw with the evaluation and not a fundamental issue with the attack. In addition, the present authors state that the “clean-label” case that they consider is a “new type of attack” (line 72); but Suciu et al. also consider the clean-label case, though they do not explicitly use that phrase. The present authors also say that this is the first time that “clean-label poisoning in the end-to-end training scenario” has been studied, but doesn’t Suciu et al. do that too (with the caveat about mini-batching)? Suciu et al. also performs a more comprehensive evaluation of data poisoning attacks and defenses, which the current paper does not consider. The main benefit of the current work is in the additional analysis of the image/NN poisoning setting, but the change from L_\inf to L2 norm makes it hard to compare (for example, Suciu et al. report successful poisoning attempts against CNNs without using watermarking; I’m not sure if that’s because of the change in norm, or something else). Overall, the paper is well-written and informative but seems to have a high degree of overlap with Suciu et al.; it could be strengthened by a better comparison with that work. Other comments: 1) I’m a bit skeptical of the experiments showing that end-to-end training is significantly harder than the transfer learning setting, because the mode architecture and dataset are different between those experiments. Is there a more comparable setting? 2) In Section 3.1 (one-shot kill attack), the empirical comparison to Koh and Liang (2017) is somewhat misleading. That paper uses the L_\inf norm to measure similarity in input space, whereas here the authors use the L2 norm. I’m not sure how to think about this: are the attack images used here more or less similar than those used in Koh and Liang? At any rate, it is not an apples-to-apples comparison, as the current writing suggests. Minor comments: 3) Lines 125-126: Isn’t the L2 norm the same as the Frobenius norm? I’m confused as to why you’re using two different terms for the same quantity. 4) Algorithm 1: Please specify lambda 5) There’s a missing word in the first sentence of the abstract: “Data poisoning an attack”... 6) Line 59 of the supplement has a missing reference

Reviewer 3



This paper study a targeted poisoning attack on neural nets. It proposed a clean label attack which is involved in creating new examples with adversarial noise that will be classified correctly by the classifier, however, it will cause the network to misclassify examples in which the adversarial noise was constructed from. Exploring the robustness of NN to different attack is an important research direction and the main idea of this attack is interesting, however, I have some concerns reg. this work. According to the authors: "...Our strategy assumes that the attacker has no knowledge of the training data but has knowledge of the model and its parameters...". Although white box attacks are interesting to start with, in order for this attack to be realistic, I would expect the authors to run black-box attacks as well. Assuming the authors has access to the model and its gradients but not to the training data is somewhat weird setting. It seems from the experiments that the authors use only binary classification, is there a specific reason for that or maybe I did not understand it correctly? In general, I find the experimental results not sufficient, I would expect to see this experiments on large-scale multiclass problems when proposing a new attack. Reg, defenses, it is highly important to see how this attack will work under popular defenses. The authors wrote: "...As long as the data matrix is full-rank (no duplicates), the system of equations that needs to be solved to find the weight vector is under-determined and as a consequence has multiple solutions: overfitting on all of the training data is certain to occur." do the authors have evidence to support that? In section 4.2.1, it is not clear how the authors generated the poison examples. Did you use the same base class for all examples? did you use a single base for each poison? Lastly, the authors pointed out that this approach is akin to adversarial training. The main difference between the proposed approach to AT is that in AT we draw adv noise from various instances rather than one. Since adv. training might be the most efficient way to deal with adv. examples (for now), it would be very interesting nice to analyze this side effect, if it also exists in AT. Notations: in section 2, the authors should properly define the notations, I find some missing ones, like t, b ----- RESPONSE ----- Since this specific attack was conducted using something very similar to adv. training technique (the authors also pointed it out in the conclusion section), I would expect to see some black-box/grey-box settings. Additionally, I would expect to see some detailed analysis of this phenomena (when will it work/when it won't, how many examples do I need to make it happen, what is the acc. of an end-to-end model on CIFAR10 after fine-tune with the poisoned examples?). As I see it, this is closely related to adv. training (which is one of the critical components in defending against adv. examples), hence it is crucial for the comunity to understand if this attack will affect models that have been trained like that. On the other hand, this might be out of the scope of this paper.